# A Tracer Study on Child Participation in Child Councillor Programmes Aimed towards Development of the Child Friendly Cities Initiative

**DOI:** 10.3390/children10040732

**Published:** 2023-04-15

**Authors:** Amelia Alias, Nurfaradilla Mohamad Nasri, Mohd Mahzan Awang

**Affiliations:** Faculty of Education, Universiti Kebangsaan Malaysia, Bangi 43600, Selangor, Malaysia

**Keywords:** child participation, Lundy’s model, child councillors, child-friendly cities, CFCI

## Abstract

The Child Friendly Cities Initiative introduced by UNICEF aims to help local governments realise child rights by utilising the UN Convention on the Rights of the Child as its foundation. Using Lundy’s model of child participation, which focuses on spaces, voice, audience, and influence, this study assesses young people’s participation as child councillors in two programmes in one city in Malaysia. Ten young people who were former child councillors in one state in Malaysia participated in this study. This study employed thematic analysis in analysing the data obtained using focus groups. Based on the data presented, it was clear that adult (the responsible party) understanding of meaningful child participation is still weak. This study offers substantial contributions to the limited body of literature on child participation in Malaysia by focusing on the difficulties of former child councillors in engaging in meaningful participation. Thus, more efforts (for example, by using participatory methods) are needed to educate the responsible party on the importance of addressing the power dynamic between children and adults so that children can participate effectively in decision-making processes.

## 1. Introduction

In 1996, as the world was becoming more urban and decentralised, the United Nation Children’s Fund (UNICEF) and the UN Environment Programme initiated a project to put into action the decisions made at the second United Nations Conference on Human Settlements (Habitat II), which stated that children’s happiness is the ultimate indicator of a healthy habitat, a democratic society, and good governance. A “Child Friendly City”, as the name implies, is a city, town, community, or any system of local governance that is dedicated to protecting and promoting children’s rights in accordance with the United Nation Convention on the Rights of the Child (UNCRC). The Child Friendly Cities Initiative (CFCI) is one in which children’s ideas, opinions, concerns, and rights are central to planning, decision making, and policy implementation [1]. 

Recent years have seen an increase in mayors and municipal governments advocating for the rights of children, youth, and other vulnerable members of their communities. By promoting children and youth participation in local decision making and the satisfaction of their rights and needs, the CFCI has had a positive impact in nearly 3000 cities throughout the world [1]. Malaysia is not an exception when it comes to engaging in this project. To date, there have been three (3) cities in Malaysia that have signed a memorandum of understanding with UNICEF Malaysia to be part of the CFCI. Since Malaysia’s CFCI involvement is just beginning, there is a lot to learn to ensure everything goes off without a hitch and has a positive impact. However, there is a lack of reference and evidence in the Malaysian context that might serve as a learning resource for all parties involved. Although Malaysia ratified the UNCRC in 1995, there are still many gaps in the research undertaken. After conducting a literature search, research on children’s rights in Malaysia was found to be more focused on legislation and child protection. It was found that studies involving the participation or voice of children and adolescents in Malaysia are still limited. This prospective qualitative analysis was designed to evaluate completed workshops and programmes for young people’s participant satisfaction and the challenges and hurdles they experienced during their participation. 

## 2. Literature Review

### 2.1. Child Friendly Cities Initiative and Its Challenges

A child-friendly city, according to Brown [2], respects children’s rights; is safe; has space for play; nurtures child–caregiver interactions and independent mobility; and includes children in urban policymaking and design processes. In the ongoing research on child-friendly cities, urban public spaces, the roles of housing, school, transportation, community networks, play and green space, and governance as essential prerequisites for urban living with children are investigated and highlighted [3]. For example, Saragih and Subroto [4] look at how female students apply strategies to overcome male domination at school playgrounds, which represents politics in an urban public space. Their study demonstrates that limited school property does influence the well-being of students. Nunma and Kanki [5] investigate the features of playgrounds under flyovers and motorways in Bangkok, as well as how children feel about using these areas as playgrounds. The results of Nunma and Kanki’s research shed light on how these children play, and these findings can be used by the authorities to improve public playgrounds in areas with lots of people. 

Much discussion has also taken place on the challenges in implementing CFCI. Wahyuni and colleagues [6] highlight a lack of collaboration due to the failure of stakeholders to reach a mutual understanding, a scenario caused by sectoral egos, a lack of mutual understanding between stakeholders, and a lack of involvement of non-state stakeholders. He Lee and Na Lee [7] examine children’s awareness of their rights and their satisfaction in life in the urban community environment in Korea. Zhou and colleagues [8] measure the child friendliness of one city in China and found that the current level of child friendliness in China is low. Soeharnis and Laksomono [9] highlight the problems faced by parents in understanding child-friendly policies in West Java, Indonesia. They stress the importance of fully utilising and exercising communication elements (an experienced and skilled speaker, clear and understandable message, correct and suitable channels) to effectively promote the policies. 

One common problem encountered with respect to the CFCI is insufficient understanding of meaningful child participation by adults [2,10,11]. There were situations wherein facilitators in Italy’s Sustainable Cities for Girls and Boys initiative functioned more as educators, and children complained that the adults did not consider their ideas [10]. Mahara [12] noted that there are instances of gender-based discrimination in the classroom in Greater Amman Municipality and that the municipal council does not consider the perspectives of children. Barra Mansa, Brazil, has a budgetary council; however, only children who are enrolled in school are eligible to participate, erasing the perspectives of children who are marginalised, displaced, or unable to attend school. Children in Italy express their belief that their educators select “the best ones” to participate in local governance [13]. This is also the case in Kawasaki, Japan, where participation in local governance is restricted to those who have a strong academic record [11]. 

These demonstrate that in some areas, the CFCI is not being implemented in accordance with the key criteria, which is to ensure that children’s ideas, opinions, concerns, and rights are at the forefront of planning, decision making, and policy implementation. Children were excluded from certain decisions and consequently they developed scepticism towards adults, insinuating that some adults would use the CFCI for only name and recognition. 

### 2.2. Challenges in Child Participation

One of the vital components of CFCI is child participation [1]. Access to information, a voice in decision making, and the ability to influence positive change in their family, school, community, and online all fall under the umbrella of “child participation.” Article 12 of the UNCRC is pivotal because it ensures children’s freedom to express their views freely in all situations that affect them, and that their ideas must be given sufficient weight in accordance with their age and maturity. Articles 5, 9, 13, 15, 17, and 29 of the convention have also been extensively conceptualised under the term “participation.”

Participation is a basic right that benefits children and young people. Children and young people’s active participation can inform legislation, policies, budget allocations, and service, leading to better health, education, and family life. Participation boosts self-esteem and self-belief by helping children gain skills, knowledge, and confidence. Participation also increases children’s safety by providing them with information, encouraging them to voice concerns, and presenting safe and accessible strategies for resolving violence and abuse are key to protecting them. Participation promotes civic engagement. Children can build peaceful, democratic, human-rights-respecting societies through discussion. Participation promotes accountability by holding governments and others responsible for their actions [14]. 

Although involving or consulting with children has numerous benefits, it can be difficult to put into practice. The application of Article 12 often fails because adults doubt children’s decision-making abilities and worry that allowing children more control will undermine their authority; in sum, it is related to the power relationship between children and adults. Implementing and promoting Article 12 in practice is even harder in Southeast and East Asia, where adults usually have more power over children and childhood is seen as a time of dependence instead of innocence [14]. Regarding this, Malaysian children are unlikely to express their views openly on issues affecting their lives because of cultural norms and expectations. They may feel shy, insecure, or as if their opinions are unworthy of consideration. The family environment has a significant impact on child development. A lack of parental support and understanding of a child’s needs and aspirations leads to them being more submissive and incapable of expressing their opinions. Parents frequently exclude children from family decision-making processes because they believe they lack the maturity to understand such decisions.

According to Malson and Bolzan [15], the concept of “child participation” as a “right” for children to have their opinions heard is not consistently or meaningfully operationalised in the Asia-Pacific area. In Asian-Pacific culture, the conception of children “taking part in” or “joining in with others’ activities” prevailed as the dominant construction of child participation for a long time. It is frequently related to notions of obligation, and traditionally, it has been given precedence above the rights of individuals to prioritise responsibilities owed to one’s family and community. The construction of child participation, which involves the sharing or transferring of power between adults and children, has been acknowledged to be crucial and potentially transformative of adult–child relations; yet, there are issues with acceptability among adults in Asian countries.

Promoting meaningful participation takes time and demands significant effort, not just from children, but also from adults. It is necessary for adults to overcome resistance to participation, make an effort to address social norms, dissolve their power and decrease control, learn how to negotiate with children, understand the social construction of childhood, be willing to listen with purpose, and be willing to learn from children [16,17,18,19]. 

There are various approaches that could be adopted in addressing the issue of the power relationship between children and adults. West [16] suggested adults use a participatory training method in engaging with children or adolescents. He argued that adults would be able to see and understand children’s competence (i.e., skills, comprehension, and knowledge) if they used the participatory method, which would lead to them beginning to considerably change their perceptions of children and childhood. Using a fun and playful approach in dealing with children also helps lessen the power relationship between children and adults as well as strengthening children decision-making skills [20]. Wynes [21] asserted the importance of intergenerational dialogue and interdependencies between adults and children in the development of meaningful child participation. 

Participation, according to Lundy, entails going beyond simple listening to what Harmon [19] terms “listening with purpose”. A significant obstacle in using a child’s rights approach to voice is ensuring that adults engage in purposeful listening. Purposeful listening enables the adult to be receptive to what is expressed and, when necessary, to act for the children’s benefit. The task is to be receptive to what children have to say and to guarantee that children believe they can affect the world around them. This also relates to a focus on adopting the reflective practice and experiential learning related to children’s participation. This includes watching how adults interact with each other, how adults interact with children, and how children interact with each other.

### 2.3. Lundy’s Model of Child Participation

As suggested by Lundy [17], promoting meaningful child participation involves four elements: (1) safe space—creating a safe environment and opportunities for children to access information, share their views, and make decisions related to issues they are grappling with; (2) voice—allowing for them to freely express their own views in a form of choice; (3) audience—hearing their point of view; and (4) influence—considering their point of view and communicating decisions.

Children and adolescents can participate in activities and decision making in one of three ways: consultative, collaborative, or child- or adolescent-led [22]. Consultative participation is adult-initiated, wherein the activities are led and managed by adults. This type of participation recognises that children and adolescents have knowledge, experience, and viewpoints that can inform decision making, but it does not transfer decision-making duties to them. Collaborative participation is also an adult-led initiative. Collaborative participation involves working with children and adolescents and allowing them to share the responsibility of decision making with adults and impacts programme process and outcomes. Finally, child- or adolescent-led participation is when a children or adolescents can make space and opportunities for themselves to start their own plans. This kind of participation gives them more power and influence, such as when they start and run their own organisations or find problems in their communities that they wish to address. In general, the role of adults in child- or adolescent-led participation is to allow them to reach their own goals by giving them the space, information, guidance, contacts, resources, and support they need [22].

Lundy’s model has been used in several studies to ensure that all children’s views are valued and respected [17]. For example, Harmon [19] used Lundy’s model to assess her decision to engage children as co-researchers in her study. Harmon concluded that employing Lundy’s model is a process comprising a series of steps that are developed over time, as trust is built with the children. The other study that used Lundy’s model in child welfare practice was conducted by Kennan and colleagues [23].

By using Lundy’s model of child participation in addition to participants’ reflections on their journeys as child councillors, this study raises several questions. These questions include whether adults, especially those from the CFCI local coordination unit, really understand the meaning of child participation, whether the activities conducted during the programme to select child councillors are child-friendly enough, and what adults could learn from this reflection. Additionally, this study asks what could be done to sensitise adults to child participation? To provide answers to these questions, we have condensed them into the following research question: What are young people’s views on the CFCI local coordination unit’s understanding of meaningful child participation?

The reflection method is used in this study as it is considered an important element for learning [24,25]. Reflection is seen as essential for connecting theory and practice [26]. Boud and colleagues [27] identified reflection as a process that looks back on experience to obtain new perspectives and inform future behaviour.

To assist participants in understanding the context of this study, two (2) programmes in which they participated were chosen as a reflection mechanism. The first author acted as an observer during both programmes.

## 3. Research Context

### 3.1. The Outreach Workshop Session

In September 2019, a series of outreach workshop sessions were conducted with two objectives. The first objective was to educate children aged 7 to 18 years old about child rights and child participation, and the second aim was to set up a child council. The objective of creating the child council was to build a platform wherein all children and young people felt collectively empowered and could work towards creating meaningful child participation practices. The use of children councils is by far the most prevalent approach to including children in the decision-making process [13]. The purpose of these councils is to encourage children to participate in the management of the city, as well as to assist children in developing a feeling of belonging in the community and respect for common values and a history of common experiences. With this strategy, children and young people are given a voice in the local government, allowing them to discuss the goals that are important to them as citizens and to collaborate with many institutions. These kinds of programmes excite local governments quite a bit since they see them as an opportunity to involve the city’s youth and cultivate a new sense of civic engagement [13]. In addition, the councils become advocates for a variety of programmes, such as those dealing with the protection of children, the rights of children, and any other projects relating to children.

In total, around 150 children from more than 20 primary and secondary schools from one city in Malaysia participated in these sessions. Every outreach workshop started with a fun, playful approach and active warm up session (energizers) to establish a friendly relationship between participants and adult facilitators. It was envisaged that this warm up session would serve to lessen the power imbalances between participants and adult facilitators. In this energizer session, participants were divided into groups where they were asked to communicate and talk among themselves to get to know more about their new friends, i.e., their hobbies, favourite food, and school.

All activities during the outreach workshops were conducted using the participatory work group method. The reason for this is that the facilitators wanted to see how participants conducted their group discussions on the topic given. It was important to observe how the children expressed their views on serious subjects and in formal situations. Each group was given a large mahjong paper so they could write and organize their ideas systematically. Then, each of the ideas was presented to all participants and adult facilitators. This was followed by a question-and-answer session.

The outreach workshops were conducted using a consultative level of participation approach where, at this stage, all the activities were initiated and led and managed by the adult. Adult in this context refers to the CFCI local coordination unit and the appointed child advocates from a non-governmental organisation (NGO). However, wherever possible, opportunities were given to different children to lead discussions and to present their group’s views. The first author was informed that the main reason for using this approach was because this is the first time the children were introduced to the idea of child rights and child participation.

The outreach workshops were designed to be informative and educational. During the workshops, participants were introduced to the concepts and ideas of child rights (Convention on the Rights of the Child 1989). Participants were divided into groups and given a set of colourful cards that summarised information on child rights (Figure 1).

The other activity involved using child rights picture cards and getting children to group them into the four categories of rights (survival, development, protection, and participation). Children had to discuss and agree as a group (consensus building) to categorise the rights on each of the cards under the four categories, starting with survival rights (Figure 2).

As for development rights, participants were asked to write on leaf-shaped papers what they need to have to grow and develop well, and they then had to paste it on the Development Tree under the key domains of development (physical, socio-emotional, language and communication, and cognitive development) (Figure 3).

For protection rights, participants had to brainstorm what kinds of issues children need to be protected from and who they can contact if they need help. Finally, in the participation rights activity, participants were given case studies or stories and they were required to place these on the Ladder of Participation, a concept proposed by Harts [28], with 1 being the lowest level (manipulation) and 8 being the highest (initiated by child and agreed with adults) (Figure 4).

The other activity conducted was a group discussion on six (6) various themes: cleanliness, public amenities, vandalism, bullying, child participation, and child-friendly spaces. During this session, participants were divided into groups to further discuss the above issues. This included conducting the “Why Analysis” in depth based on the template given and proposed solutions, i.e., what individuals, communities, and the government can do. The activities were followed by a group presentation from the participants (Figure 5 and Figure 6).

As mentioned earlier, one of the objectives of the outreach workshops was to select several children to be in the child council. The selection was performed by the adult facilitators based on criteria such as children’s ability to work in a group and contribute ideas, good interaction with audience, active participation and interactions, confidence in public speaking, interesting observations of his or her neighbourhood, enthusiasm about being in the child council, proposals of interesting and creative ideas, and confidence. Children were selected regardless of their age, gender, and socio-economic statuses. Finally, 32 children were selected, 20 of which were girls and 12 of which were boys. Fewer boys were selected as they participated less than girls. At the time of the study, seven (7) of them were between 10 and 12 years old, 11 were between 13 and 15 years old, and 11 were 16 between 17 years old. However, no children with physical disabilities were selected to be in the child council as they participated very little during the outreach workshop sessions.

### 3.2. The Child Friendly City Initiative Conference

In contrast to the outreach workshop sessions, the two-day conference that took place in November 2019 was organised in such a way as to encourage collaborative involvement and was designed to be attended by a large number of children and adolescents.

During this conference, the 32 children who were selected as the child councillors were given various tasks, for example, as an emcee, forum moderator, forum panellist, member of the social media team, member of the admin team, member of the energizer team, and group facilitator. The hall was set up in a child-friendly way, providing chill space with bean bags and areas where the children could leave their thoughts and ideas about their needs and child-friendly city. Almost 150 children, including children with disabilities and children from marginalised groups (indigenous), participated in this conference. During this conference, children were divided into 10 smaller groups for group activities.

The first group activity conducted was a group discussion on the six (6) main issues (cleanliness, public amenities, vandalism, bullying, child participation, child-friendly spaces), which were discussed during the outreach workshop sessions. In this session, 10 child councillors were given a task as the child facilitators with assistance from the adult facilitator (Figure 7).

The second activity involved a child moderator and two child panellists in two forums named “Towards Active Child Participation” and “No Child Left Behind.” The remaining panellists were adults. The third and last exercise was the construction of a 3D city model utilising low-cost materials such as plastic bottles, cardboard, and old paper (Figure 8). Interspersed between each activity were energizers and performances by the other children. The culmination of this conference occurred when the mayor officially appointed the 32 children as child councillors for the period of 2019–2021.

## 4. Materials and Methods

### 4.1. Sampling Approach

Of the 32 children who had previously served as child councillors from 2019 to 2021, ten (10) 19–20-year-olds were recruited as they spent most of their time participating and organising child-led activities.

### 4.2. Research Methods

A qualitative approach was used to generate detail data [29,30]. Two (2) face-to-face focus groups were conducted, which took about two hours for each session. The participants were given a report on the two programmes described above before the discussion in the focus groups. The purpose of the report was to serve as a tool to assist them in recalling and reflecting on their involvement during the programmes. In addition, the researcher sometimes assisted participants in recalling what took place during the outreach workshop sessions and the conference, after which they were questioned about pertinent information.

Despite the rigorous preparation of the research objectives, questions, and methodology, the ability to extract the richest data typically hinges on the relationships created and maintained, as well as the approaches the researcher employs while dealing with study participants. An example of strengthening relationships is seen in the researcher’s involvement in most of the activities with child councillors after they were formally appointed. This helps in having a deeper understanding of the issues or challenges faced by them.

The focus group interview guideline is based on Lundy’s Child Participation model, as shown in Table 1.

The focus group sessions were audio-recorded with the participants’ consent. To ensure the study’s credibility, consistency, and transferability, the member check technique was conducted by providing the participants with the verbatim transcripts. The participants were given a total of fourteen days to read through the verbatim transcripts, and they were given the opportunity to ask inquiries. The consent form was signed by everyone who took part in the current study, and all of them indicated that they were satisfied with the contents of the verbatim transcript.

### 4.3. Data Analysis

Thematic analysis was used when analysing the data collected during focus groups. This study utilised a hybrid approach to coding that combined inductive and deductive methods [31]. With this approach, we managed to obtain themes derived not only from the participants’ perspectives (inductively), but also from our knowledge of the theoretical aspects of the Lundy model (deductive). Space (opportunity), voice (facilitated), audience (listened to), and influence (actioned upon) were the themes’ names, which encapsulated the essence of the theme’s focus. Subthemes were determined using the topic (descriptive) coding and in vivo coding methodologies. In in vivo coding, the actual words that participants mentioned were used as the subthemes. In descriptive coding, a code was assigned to a passage based on the topic discussed (in the focus group discussion) or what was written [32,33].

## 5. Findings

Table 2 summarises the key themes and subthemes that were identified during analysis of the data. The findings are presented based on the four (4) components.

### 5.1. Theme 1: Space (Opportunity)

#### 5.1.1. Subtheme 1: Lack of Opportunity for Children to Make Decisions

As mentioned earlier, the conference was meant to be a collaborative participation. To reflect that, participants were asked whether they were consulted before the conference.

“*As far as I am concerned, the things were set in stone, right? We just did the labour and was a bit rush.*”B4

“*It was already planned and yea always last-minute decision. We just have to do it. The idea of the conference supposed to be child-led (child organise), but it was all adult.*”B5

According to UNICEF’s Conceptual Framework for the Measurement of Adolescent Participation [22], collaborative participation provides the opportunity for shared decision making with adults, and for adolescents to influence both the process and the outcomes in any given programme. However, this was not exercised during the conference.

#### 5.1.2. Subtheme 2: Lack of Opportunity to Socialise

During the focus group session, participants were asked about the space and the setup of the place during outreach and conference session. All participants agreed that the activities were carried out in a very suitable place that was very spacious, safe, and comfortable. However, when asked about the usage of the chill space that was provided during the conference, this was the answer from one of the participants:

“*I think the problem is that the children weren’t given a time to be there (at the chill space). What I can remember, only me and the other child councillor was seating on the beanbag. The flow of the conference is very formal like every time has something happening, they must stay at their place.*”B1

This was supported by the other participant:

“*The entire vibe did not give many opportunities for children to socialize or have fun during the conference.*”G1

#### 5.1.3. Subtheme 3: Unsuitable Programme for Children

All (10/10) participants concurred that the activities during the outreach workshops were arranged and conducted in a child-friendly way; however, that was not the case during the conference, as reflected by two participants above. The below was the reason highlighted by the other participant:

“*It has to be integrated into the programme flow itself. Like a chill session, like purely to have fun, to socialise. You know when we get into the group work, everyone was kind of shy, there wasn’t comfort in giving opinions. If you want to have a chill space, like the other participants mentioned, it did not give many opportunities for children to socialize and have fun. It’s like being integrated into the flow of the programme itself, everything being so formal then I think the transition to making them more comfortable will be difficult.*”B2

The above participants linked the failure of the conference as a child-friendly programme with the overly formal atmosphere and program arranged by the CFCI local coordination unit, which failed to engage with children. One example of this was seen during the panel discussion (forum) session. This is echoed in the below excerpt:

“*Well, what is the point of that forum? Is it really for the kids or for the adults? Because right now like I cannot imagine where a kid would say I really would like to go to a forum about child friendly cities. I get an idea to spark a discussion, but I think that would have to be in a smaller group. I cannot see a bunch of kids sat in rows listening to a bunch of adults and talk about things that they don’t understand. The general style of the forum is you are talking to the people on the panel, but I feel that with kids you need to talk to them.*”B3

This was further explained by the other participants:

“*It is a good topic [the forum], but for children to understand, I think it’s difficult. Normally that type of discussion should be conducted among the adults, then they need to communicate to children in simple and easy to comprehend style or approach. They just talk without any slides, so difficult for children to capture what they said.*”G2

As young people who experienced childhood, both participants know the way that the panel discussion (forum) was conducted was not suitable for children. Children are known for having short attention span, especially on matters they do not understand and unable to focus on.

Meaningful involvement of children and young people in the decision-making process requires provision of an opportunity for involvement in the form of a forum in which children and young people are encouraged to voice their opinions about the conference activities. However, this was not exercised as there were no opportunities allowing child councillors to identify topics they wanted to discuss, as these were determined by adults, as reflected in Subtheme 1 (Theme 1). In addition, some of the programmes arranged during the conference were not child-friendly, and they were unable to enjoy and feel at ease, as reflected in Subtheme 2 and 3 (Theme 1).

### 5.2. Theme 2: Voice (Facilitated)

#### 5.2.1. Subtheme 1: Failure of Adults to Understand

Participants were also asked whether they thought the adults who facilitated the programmes fully understood how to work with children. These were their responses:

“*The adult facilitators (child advocators), they know how to work with kids. For child facilitators (during conference), we were not being trained. I remembered I went to children with hearing impairment group, the child facilitator does not know what to do.*”G3

“*I don’t remember talking to them (CFCI local coordination unit) as much. Oh! I remembered that our President got fever. I think due to lack of sleep and lack of rest as she was in-charged as child panellist.*”B2

“*Overall, the organiser (CFCI local coordination unit) was not kind enough, they were not child-friendly enough, because when all the very important persons (VIPs) came, all of them surrounded and there was no chance for the kids to speak to the VIPs.*”G4

As evidenced in the excerpts above, participants expressed disappointment with how some adults treated the child councillors during the conference. This shows how children struggle with the power that adults have.

#### 5.2.2. Subtheme 2: Lack of Meaningful Inclusive Participation

Participants were also questioned about whether they felt the voice of disabled children was inadequately heard at either programme. One stated:

“*No. For me like when I do energisers during conference, most of the people I interacted with most are from the disabled kids, like they wanted to say something. Although the deaf kids were assisted by two interpreters, but I feel the opportunities given to them to do so was not enough. We should have a specific session for them to understand their needs because it’s definitely going to be different from anyone else. A separate session but during the programme itself, where everyone can hear about it. I am not saying let’s put them aside from other kids but focused on them also.*”B1

The participant above has highlighted that there is a need to have a session focusing on listening to disabled children’s needs.

Voice, according to Lundy [17], is about allowing children to express their views; hence, adults involved in programmes with children should always be respectful towards children. However, participants did not feel this was accomplished, as reflected in Subtheme 1 (Theme 2). In addition, it is essential to elicit the perspectives of a wide variety of children and ensure that participation is not limited to those who can express themselves eloquently and literately. Despite being invited to the conference, participants felt that the voice of disabled children was not adequately heard, as reflected in Subtheme 2 (Theme 2).

### 5.3. Theme 3: Audience (Listened to)

#### 5.3.1. Subtheme 1: Not being Listened to

One of the themes which emerged in this study was that participants did not have the sense that the problems they highlighted during the workshop or conference were given the appropriate amount of consideration. Typical comments were as follows:

“*When they first asked me about the problems I faced at my neighbourhood, I thought they will be doing something about it. I have expectation. But as it keeps talking about it, then I know it is a waste. Because if they really want to do, they will immediately come to the place and do research and they immediate fixed it. But they were just keep talking and talking and talking.*”G1

“*I still have hope, at least when we are doing the outreach. Then when we start to talk to the CFCI coordination unit about this stuff, then they will say “Oh is it?” and that’s it! They won’t do unless you make a case.*”G2

“*During outreach and conference a bit convincing that whatever I voice out will be addressed. At first they gave us confidence. I was thinking “this is the time for me to make a change” and now, I still keep thinking when all these will be realised. Then, the authority vanished. Expectation went down.*”G3

These are some instances of problems that may develop if children are not provided with clear information. They may develop the view that their opinion or viewpoint will not be considered.

#### 5.3.2. Subtheme 2: Lack of Planning for Child Councillors

Participants were asked whether they had been involved in or informed about the plans of the CFCI coordinator unit to support them in playing a role as councillors. These are responses from two (2) of the participants, which were agreed upon by all participants:

“*If they [CFCI coordinator unit] have plans in place, they would have informed us!*”G2

“*Agree, as far as I remembered, they will only call us to be in their event, but never called us to do the planning on our path as child councillor.*”G4

One of the criteria of weak management is failure to make plans with children or initiate discussion with children on their roles as child councillors. Planning offers great advantages to children, for example, allowing them to articulate ideas, make choices and decisions, and promote their self-confidence and sense of self-control. Having a proper and well-organised plan would not only benefit children, but also the whole CFCI coordination unit since it provides a clear direction.

The audience components in Lundy’s model relate to giving children’s views “due weight”. The idea that children have a right to have those who make decisions pay attention to what they have to say is inextricably linked to the concept of giving “due weight” to their perspectives. In other words, this component deals about how adults show that they are committed to being informed and influenced by children’s views. As expressed by participants in Subtheme 1 (Theme 3), it was clear that participants did not know to whom, how, and when their views would be communicated.

Audience is also about plans put in place to support children and young people in playing a role in communicating their own views. As shown in Subtheme 2 (Theme 3), this was also not exercised by the coordinating unit.

### 5.4. Theme 4: Influence (Acted upon)

#### 5.4.1. Subtheme 1: Fail to Inform Impact on Children’s Views

Article 12 of the UNCRC uses the phrase “due weight” to describe the concept of influence. According to Lundy [17], influence is crucial: children must feel they have impact. Therefore, while encouraging children to offer their opinions, they must be informed of the audience and given feedback on how the audience might respond to their opinion. If children perceive that their opinions matter and are respected, it will foster a culture wherein their opinions are valued in proportion to their age and maturity [19].

Participants were asked as to whether they were aware of the progress made by the coordination unit on the CFCI. Similarly, participants were also queried about whether they were informed on the impact of their views.

All (10/10) of the participants reported that they were not apprised of the development of the CFCI by the CFCI coordination unit. As expected, there was also no communication or process in place to inform them on the impact of children’s views. This was based on the previous statement by participants on the failure of the coordination unit to make a proper plan for the child councillors. This was the response from one of them on this:

“*All parties i.e., the CFCI coordination unit and the child advocate organisation should involve us as well at the beginning of every project. I feel sad sometimes, when they need us, it always last minute and in rushed. Then after that they just bye bye!*”G5

The participant above voiced dissatisfaction with the way the child councillors were treated during their term, asserting that they should be part of the plans and decisions made for matters related to them.

#### 5.4.2. Subtheme 2: Lack of Update from CFCI Coordinator

Participants were also asked for their suggestions on how the CFCI coordination unit should handle the feedback on how their ideas were used and the reasons a decision was made:

“*I think they should have meeting with the child councillors once a month. It could be end of the month. Then, both parties must update each other what is their progress. Then may be the child councillors can propose project or anything. Then the following month the responsible party should report back what they have done with the project.*”G4

The excerpt above suggests the need to have a proper coordination mechanism in place to facilitate the communication of progress between the CFCI local coordination unit and the child councillors. This effort was not in place during their term as child councillors.

The influence component in Lundy’s model relates to whether children are given opportunities to evaluate the process throughout. In addition, it relates to the provision of age-appropriate and accessible feedback explaining how children’s views are used and the reasons for the decisions taken. Neither of these activities was exercised by the coordination unit, as shown in Subthemes 1 and 2 (Theme 4).

Overall, the findings as highlighted in Themes 1 to 4, which reflect Lundy’s participation model, mirror those of previous studies that have examined the issues on the implementation of the CFCI as mentioned in the Introduction. Findings have shown that participants expressed their frustration on how the supposedly collaborative “meaningful child-friendly” programme has turned out to be a tokenistic programme where children are being used as marketing material or subject matter for local good-news stories. During the conference, there was a clear absence of a method that was suitable for the target ages and which capitalised on children’s natural propensity toward play, imagination, physical activity, and exploration. The findings also demonstrate that there was a lack of appropriate planning and coordination, which resulted in the execution of every interaction being left until the very last minute, further adding to the frustration of the children. In addition, the absence of an explanation from the beginning regarding the aim of the project or programme that involved children caused the children to be confused regarding their participation.

## 6. Discussion

This study aimed to evaluate completed workshops and programmes for young people’s participant satisfaction and the challenges and hurdles they experienced during their participation. In other words, it aimed to investigate, from the perspective of former child councillors, how the CFCI local coordinating unit understands the concept of meaningful child participation.

As Le Borgne and Tisdall [34] pointed out, adults, specifically the CFCI local coordinating unit, have a considerable influence on the extent to which children are able to exercise their participation rights. The perspectives that these adults hold on children and childhood have a significant impact on whether or not these adults acknowledge, encourage, and promote children’s participation. 

Findings in all themes (space, voice, audience, and influence) highlight the necessity for adults, and particularly the CFCI coordination unit, to have an in-depth understanding of how to relinquish some of their power, work effectively with children, and place children’s voices and rights at the forefront of planning, policymaking, and the implementation of the CFCI. This is especially important given that the CFCI coordination unit is the responsible party for coordinating the programme. If adults want to collaborate with children successfully, they need to be able to build rapport with children by engaging with them as equal participants. As a direct consequence, the capacity to cultivate trust-based relationships with one another is made possible.

Additionally, adults should rethink their roles of facilitating rather than directing when working with children. Research by Horgan and colleagues [35] demonstrates that children have positive involvement experiences. These experiences involved facilitation by adults whom the children respected and with whom they had developed some sort of relationship.

The findings in Theme 1 (Space) highlight that some programmes in the conference were not child-friendly. West [16] suggested adults use participatory methods in engaging with children or young people. He argued that adults can see and understand children’s competence (i.e., skills, comprehension, and knowledge). The participatory method would lead them to considerably change their perceptions of children and childhood. Among the participatory methods that adults and children could use are community mapping, focus group discussions, transect walks, and any other activities that help adults to listen to, respect, and act on children’s voices. Creative and participatory methods may increase children’s focus and interest in the topic of discussion. In addition, as argued by Parker and colleagues [20], when interacting with children, having a fun and playful approach not only assists in reducing the power dynamic that exists between children and adults, but it can also help children develop stronger decision-making skills. All these various approaches are effective in addressing the issues of the power relationship between children and adults and can be used by the CFCI coordination unit [36,37,38]. Discussing “space” should not merely involve providing a safe space for children to voice their opinion and concerns on matters relating to them. The space provided by adults should allow children to feel relaxed, comfortable, and have access to the facilities they need.

Findings in Theme 2 (Voice) highlight a lack of meaningfully inclusive participation during the conference. The General Assembly of the United Nations 2002 placed a strong emphasis on the importance of ensuring the participation of disadvantaged and marginalised children and young people [39]. It is necessary to foster children’s and young people’s sense of self-worth and prepare them to assume responsibility for their own lives. Evidence shows that children with disabilities participate less frequently and are less involved in participating compared to children without disabilities, possibly because of perceived access and communication barriers and negative assumptions about their abilities [40,41,42]. It is crucial to involve disabled children in community-based activities, as they are in the best position to articulate what works and how things may be improved for themselves and their families.

In Themes 3 (audience) and 4 (influence), participants expressed discontent with how they were treated during their terms as child councillors, stating that they should be part of the planning and decision-making process for matters relevant to them. The purpose of the project or programme involving children must be clear from the outset, including why they are establishing the project or programme, what they anticipate getting out of it, what they want children to get out of it, and the reality of the levels and topics of children’s decision making. In addition, proper planning and coordination are also crucial to avoid last-minute executions that might further frustrate children. This finding is comparable to that discovered by Horgan and colleagues [35], who found that children’s impressions of adults’ unfavourable or dismissive attitudes functioned as a significant barrier to their current participation and future involvement.

Findings in Theme 3 (Audience) also highlight that they were not being listened to. Therefore, from the outset of the process, children must be aware of their intended audience and how their information will be handled, as this requires clear and open communication. In addition, it is imperative that adults make it abundantly clear to children that although it is impossible to guarantee particular outcomes from these processes, it is possible to establish formal communication channels from the child or young person’s point of view. This also relates to the need for adults who want to engage with children to develop the skill of purposeful listening, as suggested by Harmon [19]. Lack of receptivity to what children have said may affect children’s belief or trust in adults.

Findings in Theme 4 (Influence) suggest that a proper mechanism should be in place to keep children informed about decisions that have been made. This is part of Lundy’s meaningful child participation criteria, which emphasise that children must receive feedback after participating in various consultations.

When taken as a whole, the way in which the CFCI is currently managed makes it evident that the coordination unit needs a member or consultant who is well versed in child rights and child participation to help develop CFCI management that is both effective and respectful of children’s interests.

## 7. Conclusions

Utilising four components of Lundy’s model, i.e., space, voice, audience, and influence, helped the researcher to meaningfully capture the voice of young people on their views about their participation as child councillors. The findings indicate that children are dissatisfied with the fact that an event intended to be the result of a collaborative effort ended up being mostly coordinated by adults. In addition, the findings demonstrate that the absence of preparation and coordination causes delays in every interaction, which further agitates the children. When it comes to participating in initiatives or programmes, children are perplexed when the purpose of the endeavour is not made clear from the very beginning.

This study makes a significant contribution to the limited literature on child participation in Malaysia by focusing on the difficulty of former child councillors in engaging in meaningful participation due to the CFCI coordination unit’s inability to comprehend the concept. However, the lack of information from the CFCI coordination unit regarding their understanding of the concept of child participation limits the scope of this study. The way the conference was conducted made it abundantly evident that adults lack knowledge regarding meaningful child participation. To determine what could be done to assist adults in meaningfully coordinating activities with children, it would be advantageous to conduct additional research on the CFCI coordination unit’s perspectives on child participation. The other limitation of this study is that only 19–20-year-olds participated in this study. Although young people may have a deeper understanding of the significance of child participation, it would be advantageous to conduct interviews with much younger child councillors to determine what those children understood about child participation.

## Figures and Tables

**Figure 1 children-10-00732-f001:**
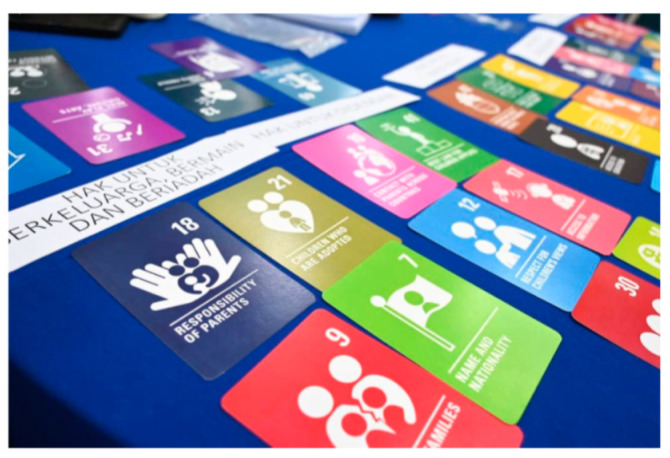
Introduction to child rights activity. Source: first author collection.

**Figure 2 children-10-00732-f002:**
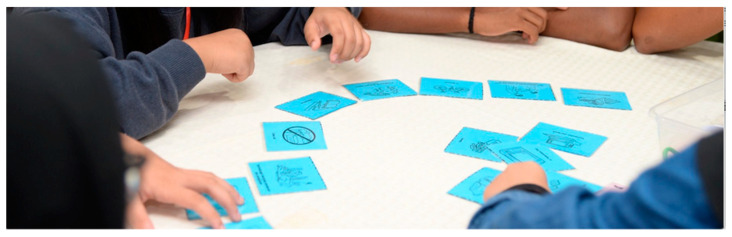
Survival rights activity. Source: first author collection.

**Figure 3 children-10-00732-f003:**
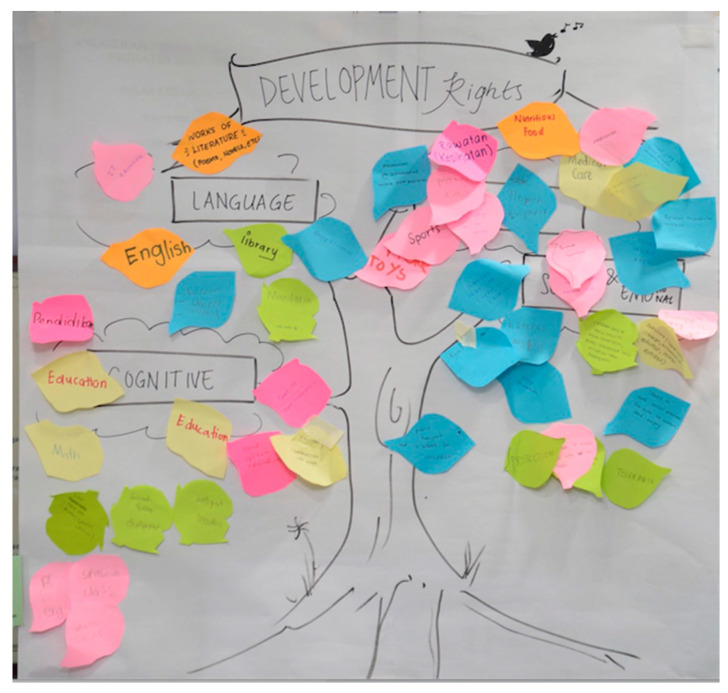
Development rights activity. Source: first author collection.

**Figure 4 children-10-00732-f004:**
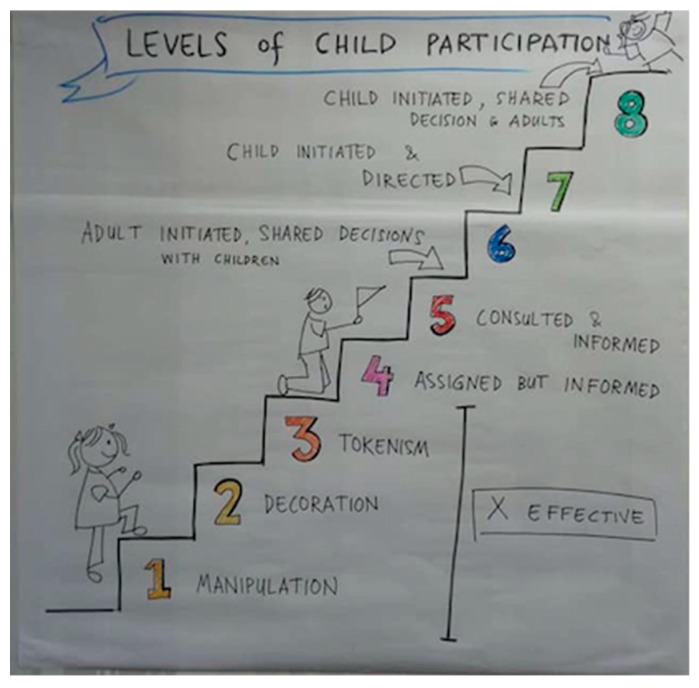
Level of child participation activity. Source: first author collection.

**Figure 5 children-10-00732-f005:**
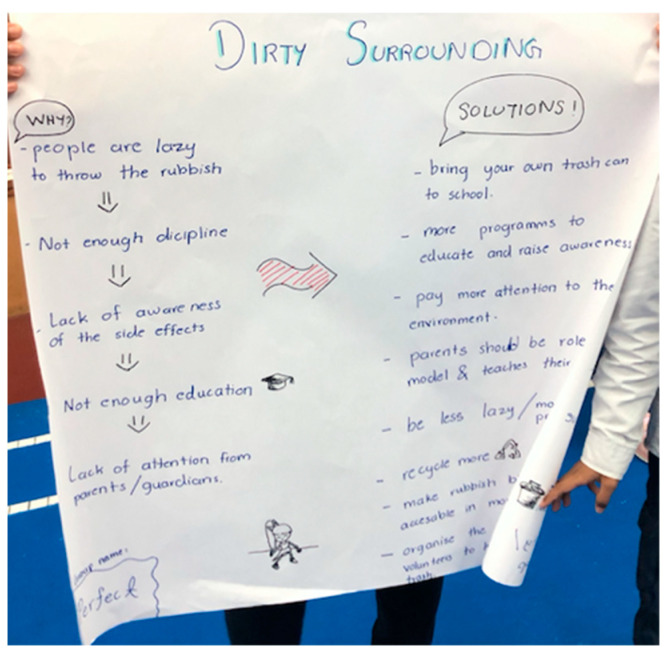
Highlighting cleanliness issues. Source: first author collection.

**Figure 6 children-10-00732-f006:**
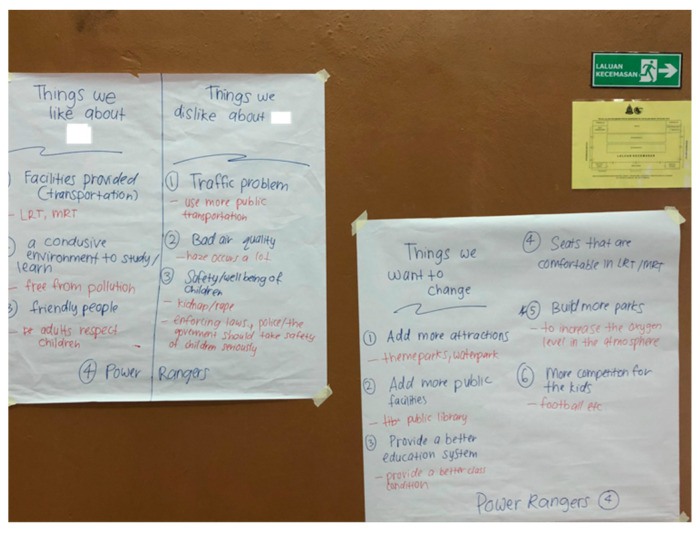
Highlighting issues: things children like/do not like about their city. Source: first author collection.

**Figure 7 children-10-00732-f007:**
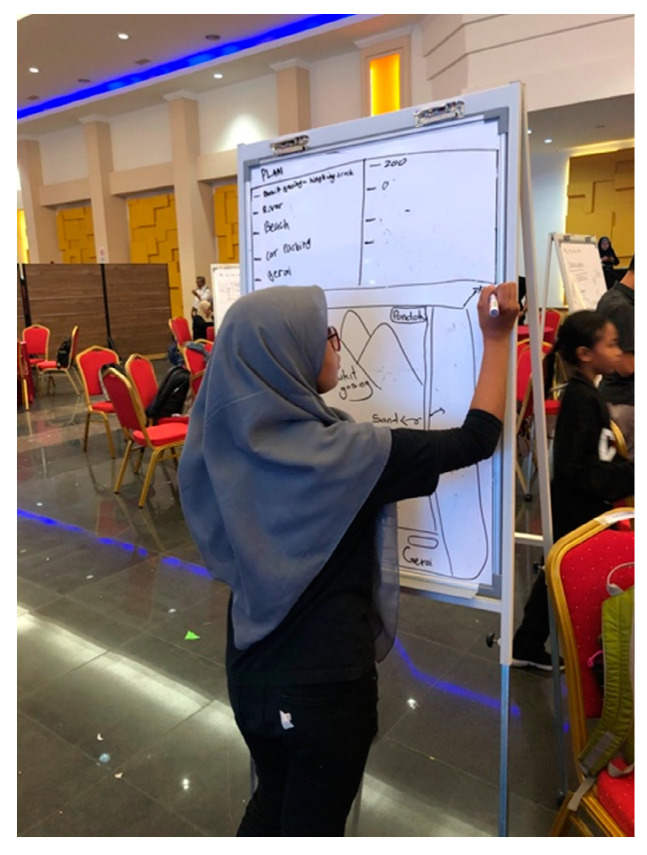
Children taking the lead as a facilitator. Source: first author collection.

**Figure 8 children-10-00732-f008:**
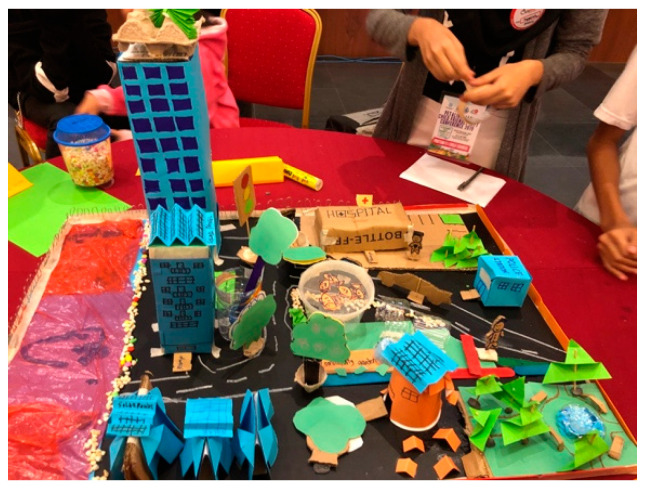
A 3D city model using low-cost materials. Source: first author collection.

**Table 1 children-10-00732-t001:** Focus group interview questions guideline.

Theme	Questions
Space	During 2019 Outreach sessions and CFCI Conference, which activities that were in line with the CRC? What about the reaction/behavior of adults as they run the program? Do you think they fully understand the rights of the child? Why? Do you feel comfortable and safe and to express your view during that time? What made you feel that? What about the space and the setup, is it child friendly? Do you think the facility given to the disable children enough to make them comfortable? Why?
Voice	During that time, do you feel that your voice or opinion will be heard? Kindly elaborate. Do you think you have been given enough autonomy to decide during that outreach session and CFCI Conference? Why? Do you think the voices of children with special needs were heard enough during the conference? Please explain more. Do you feel the topics discussed during both sessions are appropriate for children? Does the CFCI coordinator team inform you in advance about what are they going to do during the conference?
Audience	Since 2019, were you been informed of the CFCI progress by the authority? Are the things or issues that you raised during the outreach sessions has been implemented? If not, what could be the reason? What is the best way you feel the responsible party should do to communicate information about CFCI to you?
Influence	Do you think you were consulted in a variety of ways and supported to be decision-makers at all stages during that outreach and conference session? Do you have any idea where your contribution was highlighted?

**Table 2 children-10-00732-t002:** Emerging findings based on analysis. Source: data from focus group discussion.

Theme	Subtheme
Space (Opportunity)	Lack of space for children to make decisions Lack of opportunity to socialise Unsuitable programme for children
Voice (Facilitated)	Failure of adults to understand Lack of meaningful inclusive participation
Audience (Listened to)	Not being listened to Lack of planning for child councillors
Influence (Acted upon)	Failure to inform impact on children’s views Lack of update from CFCI coordinator

## Data Availability

The data presented in this study are available on request from the corresponding author.

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
