# Peer review of "A Tracer Study on Child Participation in Child Councillor Programmes Aimed towards Development of the Child Friendly Cities Initiative"

_children, 2023, doi:10.3390/children10040732_

Round 1

Reviewer 1 Report

Review of ijerph-2220898

  1. The statement of the problem of this research is not clearly seen, yet. Lines 79-81 mention about  adolescents (10-19 years) and young people (10-24 years). But in line 172, it is mentioned that the age of the participants is 7-18 years old, quite off the range mentioned in lines 79-81.
  2. Line 83-126: This manuscript must also provide more literature about the theory of child-friendly cities.

·       Cities & Health 3 (2019) 1-7 https://doi.org/10.1080/23748834.2019.1682836

·       Built Environment 36 (2010) 474-486 https://doi.org/10.2148/benv.36.4.474

·       Journal of Social Work Practice 29 (2015) 99-112 https://doi.org/10.1080/02650533.2014.993942

  1. Line 83-126: Because this manuscript also discusses about child-friendly cities for children aged 7-18, which are in their phases of primary and secondary education, then definitely additional references about child-friendly schools must be added also.

·       Journal of Asian Architecture and Building Engineering 2022  https://doi.org/10.1080/13467581.2022.2153061

·       Journal of Asian Architecture and Building Engineering 2022  https://doi.org/10.1080/13467581.2021.1928506

  1. Table 1: What is the source for this table, any data related to this table?
  2. Table 1: Please justify on how the subtheme (subthemes?) translated as the part of the results analyzed in this manuscript.
  3. Table 1: Conclusions of each subtheme in Table 1 are not vividly demonstrated and not clearly mapped to each of Lundy’s model (Space, Voice, Audience, Influence)
  4. Line 558-591: This section (6. Conclusions) has not evidently reflected the results of the analyses performed in this manuscript.
  5. Line 575-591: Please merge these sentences or short paragraphs as a one solid paragraph.

  1. Line 8: For the email address, please change the comma to dot.
  2. Line 13: Please do not start a sentence with numbers. Please change it to “Ten young…”
  3. Line 56: Please delete “In” before  “Barra Mansa”
  4. Line 69: Please change “dearth” to “lack of”
  5. Line 169: Please corroborate who is “The author”, because there are three authors in this submission.
  6. Line 169: The author acts as an..
  7. Line 171-186: Please provide the photo of the event as a proper documentary and proof. You can put it in the Supplementary Materials.
  8. Line 210-226: Please provide the photo of the event as a proper documentary and proof. You can put it in the Supplementary Materials.
  9. Line 214: …put into the four…
  10. Line 218: Please provide an illustration of the “leaf-shaped papers”. You can put it in the Supplementary Materials.
  11. Line 224-226: Please provide the samples of the quiz result and/or the workshop result. You can put them in the Supplementary Materials.
  12. Line 247-269: Please provide the photo of the event as a proper documentary and proof. You can put it in the Supplementary Materials.
  13. Line 614: 5th --> superscripted th
  14. Line 649: 2nd --> superscripted nd
  15. Line 671: Archives of Physical Medicine and Rehabilitation --> please write the name of this journal with uppercase A, P, M, and R.

Reviewer 2 Report

General comments:

This is an interesting paper that discusses an important concept. However, the introduction needs to be more to the point, with a greater emphasis on structured introduction of each idea and how it is relevant to the activity at hand and less of a general review. Focused sections describing the point of each concept and how it is relevant to the existing program would be very helpful here. 

There is considerable mixing between introduction, methods, results and discussion. The paper would benefit from reorganization, with each section having a specific purpose and clarity and no back and forth between sections. As well, focusing on brevity, avoiding passive voice is important. 

Overall, it is particularly important with any research, but particularly with qualitative research to ensure that the methods and results are very clearly described and that the themes being analyzed in the discussion are adequately captured. It is also important that the introduction is very specifically used to set up the question and justify the methods, and that the discussion demonstrates exactly what the findings add to the literature. 

Specific comments:

Line 279: in this paragraph a more specific description of the study design and detailed approach to which data was of interest, how it was collected and recorded would be helpful. It is not possible to replicate this work based on the information provided. 

Line 286: this paragraph reflects on some of the findings of the project and would be better situated in the "results" section. 

Line 301: This is only one example, but throughout the manuscript, comments like "helped with the data analysis" are more of a discussion of what worked and did not work (Discussion) than methods. Having a mixture of methods, results and discussion confuses the reader. In addition, statements like in line 309 about "drawing on..." almost belong in the introduction, which is the appropriate space to introduce and make a case for relevant methodological approaches. 

Reviewer 3 Report

I encourage the authors to add the cultural specifics of childhood in the area of interest, children's relationship to play and independent decision-making, and cooperation with adults. 

Round 2

Reviewer 1 Report

This manuscript is significantly improved. It can be accepted in the present form.

Author Response

Thank you for positive feedback. Highly appreciated

Reviewer 2 Report

Thank you for the revised submission.

General comments:

Introduction - the introduction is more organized than previously but the structure still could be improved, with each section being more to the point. The section overview should state exactly what it will be about and each paragraph have a short and focused description of the topic. The figures in some cases are nice/helpful; in others, where it is just a photo of a chart, a typed summary would be much easier for the reader to actually read. At the end of the introduction, a very short statement specifically describing the purpose of the study would be helpful. Like "this prospective qualitative analysis was designed to evaluate completed workshops for youth participant satisfaction, etc.". It is very important to reference statements throughout.

There are other aspects of the introduction that make it appear more like a textbook chapter than a journal article - for instance Table 1. Is it possible that Lundy's approach is described elsewhere? Can the authors make a short figure or table summarizing the themes they selected, leaving it to the reader to find the details of Lundy's approach?

Materials and methods:

The language still requires revision. Line 362, for example: Of the ___ participants from 2019-21, ten 19-20 year olds were recruited based on ____ characteristics. 

Line 368 is very chatty. Instead of "we decided to take a qualitative approach", something like 
A qualitative approach was used to generate detailed data (reference). 

The remainder of the methods section will benefit from an increased attention to detail with decreased wordiness overall. The project still could not be replicated from the information provided. 

Results:

The results and discussion are enmeshed. As an example, lines 419-424 are results. The following 425-428 are discussion. The remainder of the results section also has a considerable amount of discussion in it. 

Discussion: 

This section should be rewritten to present the reflections raised in the results section in an organized manner, discussing each thoughtfully based on the literature. This can be done by removing the discussion segments from the results section, merging them with the discussion, and ensuring that the whole structure reflects the thematic analysis that was completed. 

Conclusion

Typically a conclusion is a short summary of the findings of the paper, recommendations and future study opportunities. This can be achieved by moving discussion points currently in the conclusion up into the discussion section and ensuring that only the actual findings of the paper, no speculation, are presented in the conclusions, followed by identification of literature gaps and recommendations for future research or conference planning. 

Author Response

Thank you for your review and constructive comments. Please refer to the attached file for the detailed amendment. 
